# High red blood cell distribution width attenuates the effectiveness of Immune checkpoint inhibitor therapy: An exploratory study using a clinical data warehouse

Hiromi Matsumoto[1], Taichi Fukushima[2], Nobuaki Kobayashi[1]*, Yuuki Higashino[2], Suguru Muraoka[1], Yukiko Ohtsu[1], Momo Hirata[1], Kohei Somekawa[1], Ayami Kaneko[1], Ryo Nagasawa[1], Sousuke Kubo[1], Katsushi Tanaka[1], Kota Murohashi[1], Hiroaki Fujii[1], Keisuke Watanabe[1], Nobuyuki Horita[1], Yu Hara[1], Takeshi Kaneko[1]

1 Department of Pulmonology, Yokohama City University Graduate School of Medicine, Yokohama, Japan,
2 Yokohama City University School of Medicine, Yokohama, Japan

* nkobayas@yokohama-cu.ac.jp

**Data Availability Statement:** All relevant data are within the manuscript and its Supporting Information files.

## Abstract

### Background

Immune checkpoint inhibitors (ICIs) have improved outcomes in cancer treatment but are also associated with adverse events and financial burdens. Identifying accurate biomarkers is crucial for determining which patients are likely to benefit from ICIs. Current markers, such as PD-L1 expression and tumor mutation burden, exhibit limited predictive accuracy. This study utilizes a Clinical Data Warehouse (CDW) to explore the prognostic significance of novel blood-based factors, such as the neutrophil-to-lymphocyte ratio and red cell distribution width (RDW), to enhance the prediction of ICI therapy benefit.

### Methods

This retrospective study utilized an exploratory cohort from the CDW that included a variety of cancers to explore factors associated with pembrolizumab treatment duration, validated in a non-small cell lung cancer (NSCLC) patient cohort from electronic medical records (EMR) and CDW. The CDW contained anonymized data on demographics, diagnoses, medications, and tests for cancer patients treated with ICIs between 2017–2022. Logistic regression identified factors predicting $\leq 2$ or $\geq 5$ pembrolizumab doses as proxies for progression-free survival (PFS), and Receiver Operating Characteristic analysis was used to examine their predictive ability. These factors were validated by correlating doses with PFS in the EMR cohort and re-testing their significance in the CDW cohort with other ICIs. This dual approach utilized the CDW for discovery and EMR/CDW cohorts for validating prognostic biomarkers before ICI treatment.

### Results

A total of 609 cases (428 in the exploratory cohort and 181 in the validation cohort) from CDW and 44 cases from EMR were selected for study. CDW analysis revealed that

**Funding:** The author(s) received no specific funding for this work.

**Competing interests:** The authors have declared that no competing interests exist.

elevated red cell distribution width (RDW) correlated with receiving ≤2 pembrolizumab doses (p = 0.0008), with an AUC of 0.60 for predicting treatment duration. RDW's correlation with PFS (r = 0.80, p<0.0001) and its weak association with RDW (r = -0.30, p = 0.049) were confirmed in the EMR cohort. RDW also remained significant in predicting short treatment duration across various ICIs (p = 0.0081). This dual methodology verified pretreatment RDW elevation as a prognostic biomarker for shortened ICI therapy.

## Conclusion

This study suggests the utility of CDWs in identifying prognostic biomarkers for ICI therapy in cancer treatment. Elevated RDW before treatment initiation emerged as a potential biomarker of shorter therapy duration.

## Introduction

Currently, immune checkpoint inhibitors (ICIs) are achieving high treatment outcomes for various types of cancers such as malignant melanoma, lung cancer, and head and neck cancer, becoming one of the standard therapies [1–3]. Compared to traditional cytotoxic chemotherapeutics and molecular targeted therapies like tyrosine kinase inhibitors, ICIs have distinctive features, including long-lasting anti-tumor effects (a long tail effect), cancer progression early after initiation of treatment (early non-response) [4], and the occurrence of adverse events similar to autoimmune diseases mediated by the immune system [5], some of which can be severe and prognostically significant. The complexity of balancing high costs against the potential for serious adverse events elevates the identification of patients who are most likely to benefit from ICIs to a clinical and societal imperative [6]. It is crucial that our approach to the utilization of ICIs is underpinned by a robust understanding of both their therapeutic potential and the challenges they pose, to optimize patient outcomes and ensure sustainable healthcare practices.

Biomarkers such as tumor PD-L1 expression rate, microsatellite instability, and Tumor Mutation Burden are currently used to predict the effects of ICIs. However, these biomarkers have issues like insufficient predictive accuracy, variability in assessment methods (including different antibodies used for PD-L1 evaluation and potential inter-observer variability), and heterogeneity within the same tumor [7,8]. Therefore, there is a demand for the development of biomarkers that can more accurately predict the therapeutic effects of ICIs.

There have been several reports on the association between a high Neutrophil-to-Lymphocyte Ratio (NLR) and poor treatment outcomes with ICIs [9–11]. Such findings posit that routine blood test parameters might be harbingers of novel biomarkers, paving the way for a prognostic model based on non-invasive, readily available clinical data. The integration of patient demographics with laboratory findings holds the promise of a less invasive, rapidly reportable, and cost-effective prognostic toolkit. However, comprehensive analyses to substantiate these preliminary observations remain conspicuously underrepresented in current literature.

While these prognostic indicators are valuable, the methodologies for their identification have largely been confined to observational studies derived from conventional practices. There is a possibility that this approach may overlook more precise predictors that can be identified through less invasive and more convenient means. A Clinical Data Warehouse (CDW) is

characterized as a centralized repository that consolidates diverse data streams, fostering the refinement of clinical decision-making by providing strategic, domain-specific information [12]. A comprehensive analysis of clinical data within a CDW, including blood test results, could unveil previously unrecognized prognostic factors for ICI therapy. This integrative approach may yield novel insights, thereby enhancing our predictive capabilities and ultimately informing personalized therapeutic strategies for cancer patients.

Therefore, this study aims to conduct a comprehensive analysis utilizing a CDW to identify biomarkers that can presage the effectiveness of ICIs in the treatment of various malignancies prior to the initiation of therapy. The purpose extends to verifying the validity of these identified markers through conventional observational studies, thereby ascertaining their utility in clinical prognostication. This dual-faceted approach seeks to corroborate the applicability of CDW-derived prognostic markers and establish a methodological synergy between data-driven and empirical research modalities.

## Material & method

### Research overview

In this study, we conducted three major studies. In the exploratory part, factors that could predict the number of pembrolizumab doses were extracted from blood test data obtained from the CDW of pembrolizumab-treated patients before starting treatment. In the validation part 1, we tested whether the number of pembrolizumab doses is a valid surrogate marker of PFS, and whether the factors obtained in the exploratory part are also significant in real clinical data obtained from electronic medical records (EMR). In the validation part 2, we tested whether the factors obtained in the exploratory part were also significant in a population that included ICIs other than pembrolizumab (Fig 1).

This is a schema representing the overall picture of this study. The study consists of one exploratory part and two validation parts.

CDW; Clinical Data Warehouse, EMR; Electronic Medical Record, PFS; Progression Free Survival

### Clinical data warehouse (CDW)

CDW store structured, semi-structured, and unstructured data extracted from EMRs and other sources [13]. A CDW is referred to when these data are combined with multiple modalities, such as image data, prescription data, and laboratory data [14]. Yokohama City University uses SIMPRESEARCH® (4DIN Inc.), where structured data like birth dates, registered diagnoses, blood and urine test results, prescribed medications are anonymized and stored. Moreover, to ensure anonymity, birth dates and prescription dates are randomly shifted within a 30-day range for each case. On the other hand, unstructured data such as notes in EMRs and image data are not registered. As of January 2024, approximately 330,000 cases of information are stored, combining Yokohama City University Hospital and Yokohama City University Medical Center.

### Inclusion criteria

In the exploratory phase, we included patients who visited either Yokohama City University Hospital or Yokohama City University General Medical Center, had their medical information stored in the Clinical Data Warehouse (CDW), and received pembrolizumab, either as monotherapy or in combination therapy. The inclusion criteria were based on the initiation of treatment between September 2, 2017, and January 21, 2022.

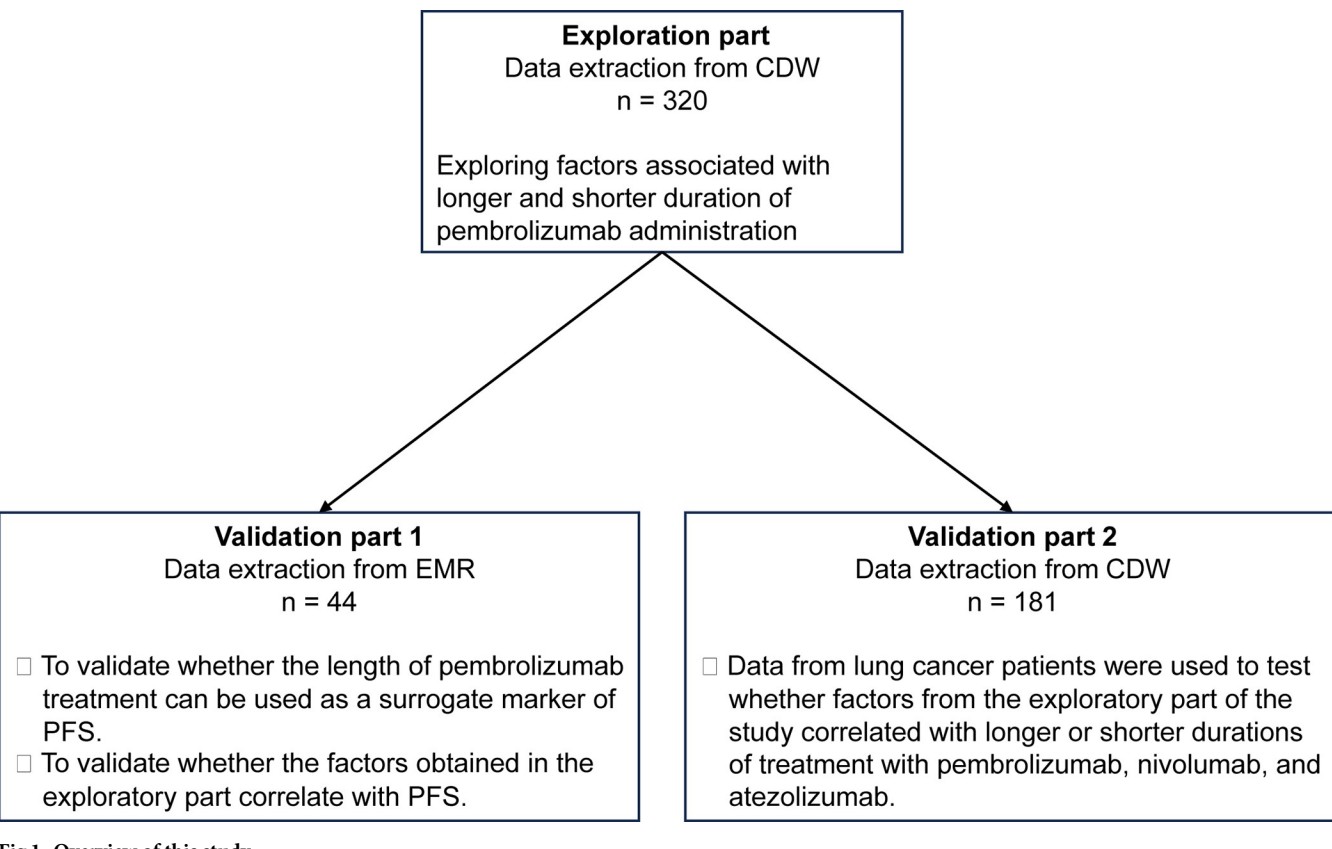

**Fig 1. Overview of this study.**

For the first validation cohort, we selected patients from the Department of Respiratory Medicine at Yokohama City University Hospital who commenced pembrolizumab treatment specifically for lung cancer between March 30, 2017, and August 20, 2021. These cases were identified through electronic medical records.

In the second validation cohort, we included patients from the two affiliated hospitals whose records in the CDW indicated that they were receiving nivolumab, pembrolizumab, or atezolizumab for non-small cell lung cancer (NSCLC). The treatment initiation date range for this cohort was aligned with that of the exploratory phase.

## Data sources

As mentioned above, information was collected from the CDW for exploration and validation 2, and from EMR for validation 1. For each case, the name of the disease, date of birth, date of pembrolizumab administration, and blood draw data immediately before administration were obtained. To obtain data, the CDW was accessed from April 6, 2023 to September 28, 2023, and the EMR was accessed from June 2, 2023 to June 30, 2023.

## Statistical analysis

A model was created to predict whether pembrolizumab was administered 2 or fewer times or 5 or more times as a surrogate marker of PFS, using the objective variable, age, and each blood draw data. The threshold for the number of doses was selected by reference to the overall distribution and the number of doses that would yield a sufficient number of cases for statistical

analysis. Because unstructured data such as EMR descriptions were not available, instead of calculating PFS directly, the number of pembrolizumab doses was used as the objective variable. Logistic regression analysis was used to determine factors related to the number of doses (short term and long term). Furthermore, to evaluate the discriminative performance of the relevant items, a Receiver Operating Characteristic (ROC) curve was constructed, and an appropriate cutoff value was determined using the Youden index[15]. Univariate regression analysis and Pearson's correlation coefficient were used to examine the relationship between the number of pembrolizumab doses and PFS from electronic medical record data. Pearson's correlation coefficient ($r$) was determined as no correlation when $|r|<0.2$, weak correlation when $0.2 \leq |r|<0.4$, moderate correlation when $0.4 \leq |r|<0.7$, and strong correlation when $0.7 \leq |r|$ [16]. Mann-Whitney's U test was used to compare patient backgrounds in each group. The software used for the analysis was python version 3.9.13, scikit-learn (ver. 1.2.1), Numpy (ver. 1.21.5), scipy (ver. 1.10.0), JMP Pro17® (JMP Statistical Discovery LLC) were used. The significance level was set at p = 0.05.

## Ethical considerations

The study protocol was approved by the Institutional Review Board of the University (approval number: B191200044). In addition, the need to obtain a research participation consent form was waived because this is a retrospective observational study.

## Result

### Exploration part

**Patient characteristics of exploration part.** From the CDW, 428 patients were identified who met the criteria. Patient background is shown in Table 1, and the median number of pembrolizumab doses was 5 (range: 1–77). There were no significant differences in age, gender, or the department in which the ICI was administered in the two groups.

Early-discontinuation group was defined as two or fewer doses, and sustained-treatment treatment was defined as five or more doses, and the above population was divided into two groups. The analysis included 320 subjects. Cases in which the period between the first dose and the last day of CDW storage was less than 12 weeks were excluded because it was impossible to determine whether they belonged to the sustained-treatment group or not. There were 99 cases in the early-discontinuation group and 221 cases in the sustained-treatment group (Table 1). There were no significant differences in age, gender, or department between the two groups.

**Identification of predictors for pembrolizumab dosing by CDW.** Age, red blood cell count (RBC), hemoglobin (Hb), hematocrit(Hct), mean corpuscular volume (MCV), mean corpuscular hemoglobin (MCH), mean corpuscular hemoglobin concentration (MCHC), red blood cell volume distribution width (RDW), white blood cell count (WBC), neutrophil count (Neu count), eosinophil count (Eo count), Basophil count (Baso count), Monocyte count (Mono count), Lymphocyte count (Lym count), Neutrophil to lymphocyte ratio (NLR), Platelet count (PLT), Mean platelet volume (MPV), Total protein (T-Pro), Albumin (Alb), Aspartate transferase (AST), Alanine aminotransferase (ALT), alkaline phosphatase (ALP), gamma-glutamyl transferase (G-GTP), total bilirubin (T-Bil), creatine kinase (CK), lactate dehydrogenase (LDH), blood urea nitrogen (BUN), creatinine (Cre), estimated Sodium (Na), potassium (K), chlorine (Cl), calcium (Ca), and C Reaction Protein (CRP) were used as explanatory variables.

Univariate logistic regression with early-discontinuation and sustained-treatment as objective variables showed that Hb (p = 0.0012), Hct (p = 0.0015), RDW (p<0.0001), WBC count

**Table 1. Patient characteristics.**

|  | Total | Early-discontinuation group | Sustained-treatment group | p-value |
|---|---|---|---|---|
| N | 428 | 99 | 221 |  |
| Age: median (range) | 40 (34–88) | 70 (34–87) | 70 (40–88) | 0.91 |
| Sex: n |  |  |  |  |
| Male | 325 | 70 | 172 | 0.21 |
| Female | 103 | 29 | 49 |  |
| Department: n |  |  |  |  |
| Respiratory Medicine | 203 | 53 | 110 | 1.0 |
| Urology | 81 | 21 | 36 |  |
| Clinical Oncology | 54 | 8 | 27 |  |
| Otorhinolaryngology | 47 | 9 | 26 |  |
| Dermatology | 28 | 5 | 15 |  |
| Hematology | 4 | 1 | 1 |  |
| Gastroenterological Surgery | 4 | 1 | 1 |  |
| Respiratory Surgery | 3 | 1 | 2 |  |
| Gynecology | 3 | 0 | 2 |  |
| Breast Surgery | 1 | 0 | 0 |  |
| Number of doses: median (range) | 5 (1–77) | 2 (1–2) | 11 (5–77) |  |

The p-values were calculated by comparing the short term and long-term groups. Mann-Whitney's U test was used to compare patient backgrounds in each group.

(p = 0.0008), neutrophil count (p = 0.0003), neutrophil-lymphocyte ratio (p = 0.0031), serum total protein (p = 0.021), serum Alb (p<0.0001), ALT (p = 0.042), CK (p = 0.0090), LDH (p = 0.0002), eGFR (p = 0.019), Na (p = 0.010), Cl (p = 0.0008) and CRP (p = 0.0009) were significant (Table 2).

Of these 15 items, 5,005 models were created with 9 items as explanatory variables, and the explanatory variables of the model with the best classification performance were selected. The model with the best classification performance among all patterns was the one in which Hb, Hct, RDW, WBC, Neut count, T-Pro, eGFR, Cl, and CRP were entered (correct response rate: 0.77, AUC: 0.67, Tables 3 and 4). For this model, the only variable that was significant was RDW (p = 0.0008). In the ROC analysis using sustained-treatment and early-discontinuation with RDW, the Area Under the Curve was 0.60, and at an RDW value of 15.5, the sensitivity for detecting the early-discontinuation administration group was 0.41 and the specificity was 0.79 (Fig 2). This value was obtained using the Youden Index (described in Methods).

## Validation part 1: Correlation between pembrolizumab administration and progression-free survival in lung cancer patients

In the first validation segment of our study, we analyzed data from 44 patients who fulfilled the inclusion criteria from the EMR. In this group, the median age was 69.5 years (range 49–83), with a predominance of male patients (68.2%). Non-squamous cell carcinoma was the most common pathology (72.7%), followed by squamous cell carcinoma (25%) (Table 5). The median PFS was 234.5 days (range 16–1933 days), and the median number of pembrolizumab doses was 7 (range: 1–68).

A strong positive correlation was observed between the number of pembrolizumab doses and PFS with a Pearson's correlation coefficient (r) of 0.80 (confidence interval: 0.67–0.89, p<0.0001), implying that a higher number of doses is associated with prolonged PFS (Fig 3). This relationship was substantiated by the regression equation, PFS days = 29.9 × number of

**Table 2. Results of univariate analysis.**

|  | p-value |
|---|---|
| Age | 0.91 |
| RBC | 0.18 |
| Hb | 0.0012* |
| Hct | 0.0015* |
| MCV | 0.078 |
| MCH | 0.059 |
| MCHC | 0.24 |
| RDW | <0.0001* |
| WBC | 0.0008* |
| Neutrophil count | 0.0003* |
| Eosinophil count | 0.39 |
| Basophil count | 0.12 |
| Monocyte count | 0.12 |
| Lymphocyte count | 0.38 |
| NLR | 0.0031* |
| Plt | 0.22 |
| MPV | 0.18 |
| T-Pro | 0.021* |
| Alb | <0.0001* |
| AST | 0.061 |
| ALT | 0.0418* |
| ALP | 0.093 |
| G-GTP | 0.065 |
| T-Bil | 0.25 |
| CK | 0.0090* |
| LDH | 0.0002* |
| BUN | 0.52 |
| Cre | 0.96 |
| eGFR | 0.019* |
| Na | 0.010* |
| K | 0.66 |
| Cl | 0.0008* |
| Ca | 0.30 |
| CRP | 0.0009* |

*; $p < 0.05$.

pembrolizumab administrations + 83.7, which further quantifies the direct association of treatment frequency with survival outcomes.

Conversely, the analysis demonstrated a weak negative correlation between RDW values and the number of pembrolizumab administrations (r = -0.30, p = 0.049), as shown in Fig 4. The corresponding regression equation, number of pembrolizumab administrations = -1.85 × RDW value + 40.0, suggests that higher RDW values might be indicative of a reduced number of pembrolizumab administrations. Although the correlation is weak, this inverse relationship could suggest a potential utility for RDW as a prognostic marker in pembrolizumab treatment. The data underscore the potential of RDW, along with other clinical parameters, to serve as a prognostic marker for treatment duration and, by extension, patient outcomes in pembrolizumab therapy.

**Table 3. Results of multivariable logistic regression.**

|  | Coefficient | Standard error | p-value |
|---|---|---|---|
| Hb | -0.40 | 0.54 | 0.46 |
| RDW | -0.23 | 0.067 | 0.00080* |
| Hct | 0.14 | 0.18 | 0.43 |
| WBC | -0.074 | 0.16 | 0.64 |
| Neu_count | -0.0055 | 0.18 | 0.98 |
| T-Pro | 0.27 | 0.23 | 0.25 |
| eGFR | -0.0085 | 0.0054 | 0.12 |
| Cl | 0.071 | 0.040 | 0.075 |
| CRP | 0.0058 | 0.035 | 0.87 |

RDW; Red blood cell Distribution Width

*; $p < 0.05$.

## Validation part 2: RDW as a predictive marker in ICI treatment outcomes

In the second part of this study, 181 patients undergoing treatment with immune checkpoint inhibitors (ICIs)—nivolumab, pembrolizumab, or atezolizumab—were evaluated using data from the Clinical Data Warehouse (CDW) of two hospital affiliates. Pembrolizumab was the predominant ICI administered (70.7%), with nivolumab (18.2%) and atezolizumab (11.1%) less frequently used. A comparison of ICI utilization patterns revealed that atezolizumab was more common in the early-discontinuation group, while nivolumab was favored in the sustained-treatment cohort.

Median RDW differed significantly between the groups: 14.8 in early discontinuation versus 14.3 in sustained treatment (p = 0.011), potentially indicating the role of RDW in treatment duration and outcomes (Table 6). The boxplot visually depicted the higher median and range of RDW values in the early discontinuation group compared to sustained treatment. Univariate regression analysis confirmed an association between RDW and treatment groups, with a coefficient of -0.19 (p = 0.0081, Fig 5). These results reinforce the notion that RDW, alongside other clinical factors, may be instrumental in predicting treatment patterns with ICIs, and could aid in tailoring patient-specific therapeutic strategies.

## Discussion

In this study, we performed three analyses utilizing the CDW and EMR. The exploratory analysis indicated a potential association between the number of pembrolizumab administrations and RDW. In the validation analysis, we found that the number of pembrolizumab administrations could serve as a surrogate marker for PFS, and that RDW correlated with PFS in real-world clinical data. Furthermore, similar analyses examining nivolumab, pembrolizumab, and atezolizumab in NSCLC also demonstrated an association between RDW and the number of ICI administrations.

**Table 4. Predictive performance of multivariate logistic regression model.**

|  | Sensitivity | Specificity | AUC |
|---|---|---|---|
| value | 0.95 | 0.35 | 0.67 |

AUC: Area Under the Curve.

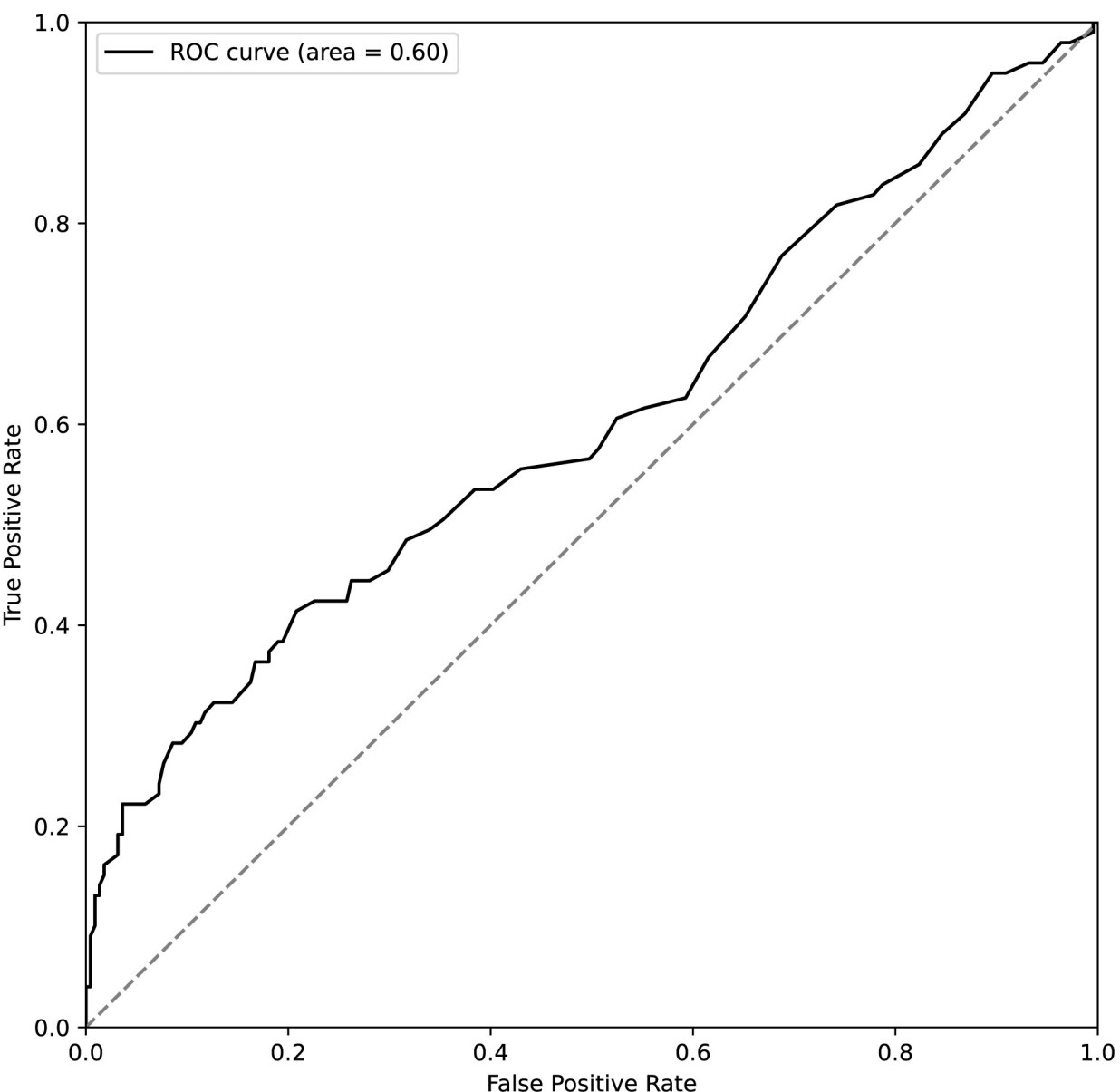

**Fig 2. ROC curve using RDW for sustained-treatment and early-discontinuation.** The ROC curve assesses the RDW's ability to discriminate between sustained-treatment and early-discontinuation of pembrolizumab, with an AUC of 0.60. At the optimal cutoff of RDW 15.5, the sensitivity is 0.41 and the specificity is 0.79 for predicting early-discontinuation. AUC; Area Under the Curve, RDW; Red blood cell Distribution Width, ROC; Receiver Operating Characteristic.

This study demonstrates the utility of CDW in identifying promising biomarkers for ICI therapy outcomes. The usefulness of clinical CDW has been shown in several studies in clinical research. In a study that assessed the risk of falls in hospitalized patients, information was gathered from EMR and clinical CDW, identifying 65 risk factors including low Body Mass Index, low blood pressure, etc.[17]. In another study on cannabis use during pregnancy, which analyzed 699 individuals, it was demonstrated that cannabis use increased the risks of preterm birth and abortion among other issues [18]. As with these studies, a major advantage of clinical CDW is the ability to collect large-scale data at low cost compared to traditional research [19].

**Table 5. Patient characteristics for validation part 1.**

| n | 44 |
| --- | --- |
| Age: median(range) | 69.5 (49–83) |
| Sex: n (%) | |
| Male | 30 (68.2%) |
| Female | 14 (31.2%) |
| Pathology: n (%) | |
| Non-Sq | 32 (72.7%) |
| Sq | 11 (25.0%) |
| N/A | 1 (2.3%) |
| PFS: median(range) | 234.5 |
| N of Pembrolizumab doses: median(range) | 7 (1–68) |
| RDW: median(range) | 14.4 (12.1–17.1) |

RDW; Red blood cell distribution width, Sq; squamous cell carcinoma.

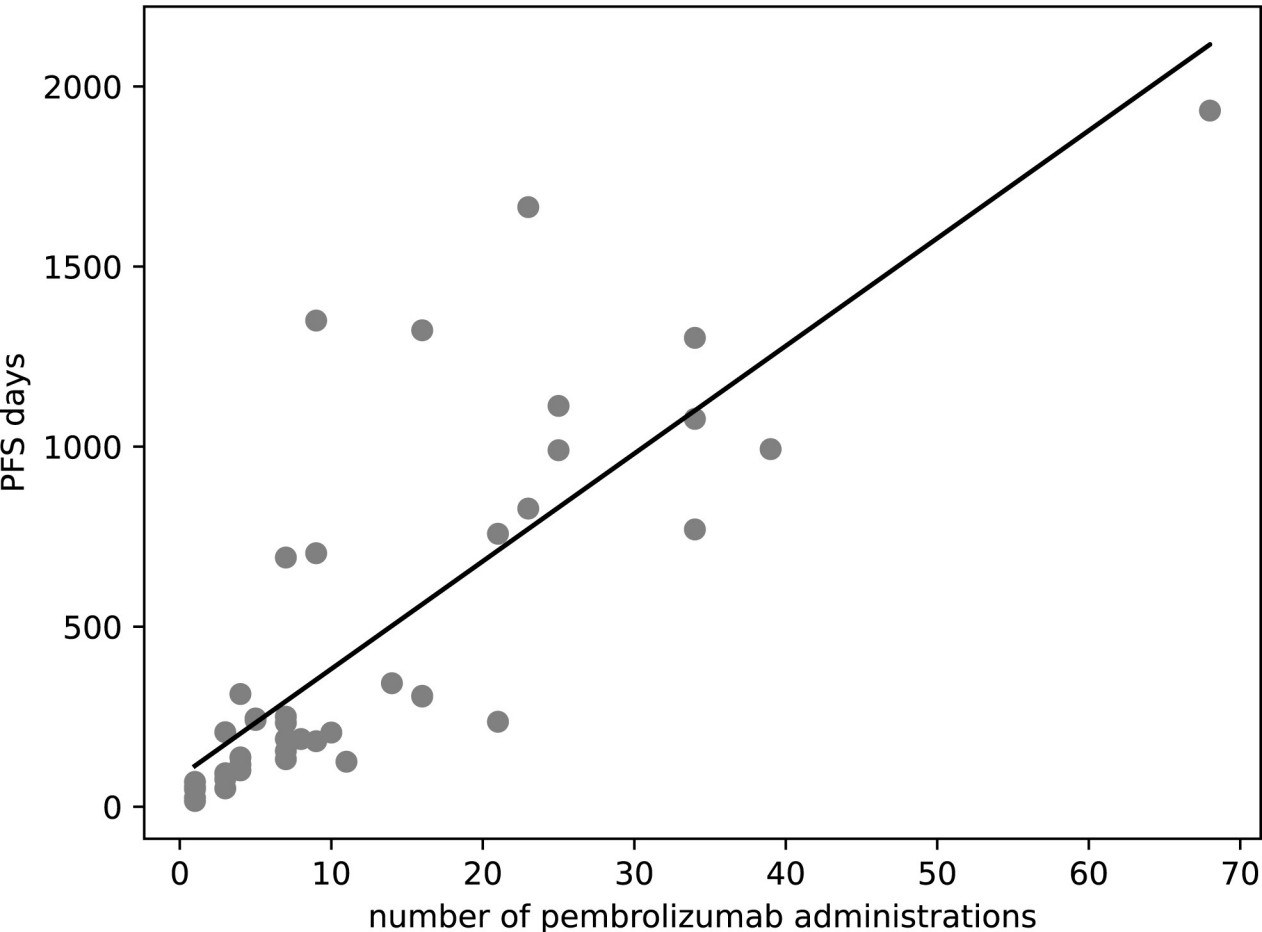

**Fig 3. Scatter plot showing the number of pembrolizumab administrations and PFS.** This scatter plot illustrates the relationship between the frequency of Pembrolizumab administrations and the duration of PFS in days. Each point represents an observed pair of the number of administrations and the corresponding PFS. Regression equation; *PFS days = 29.9 × number of pembrolizumab administrations* + 83.7, *r* = 0.80. PFS; Progression-Free Survival, *r*; Pearson's correlation coefficient.

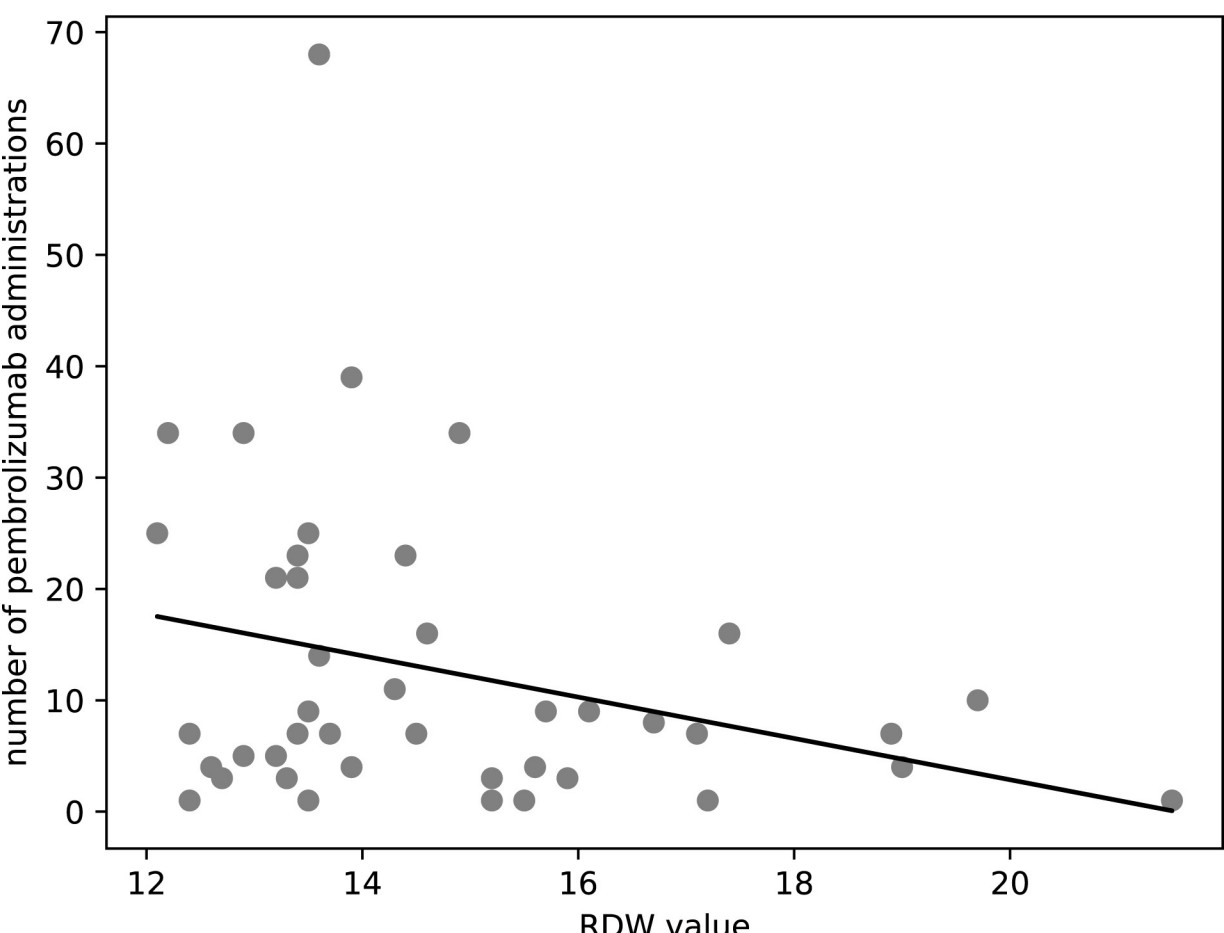

**Fig 4. Scatter plot showing RDW values and the number of pembrolizumab administrations The scatter plot shows the distribution of RDW values against the number of Pembrolizumab administrations for patients.** Each point on the plot corresponds to an individual patient's RDW value and their total pembrolizumab treatment count. Regression equation; *number of pembrolizumab administrations* = −1.85 × *RDW value* + 40.0, *r* = −0.30 RDW; Red blood cell Distribution Width, *r*; Pearson's correlation coefficient.

**Table 6. Patient characteristics for validation part 2.**

|  | Total | Early-discontinuation group | Sustained-treatment group | p value |
|---|---|---|---|---|
| n | 181 | 66 | 85 |  |
| Age: median(range) | 70 (34–87) | 70 (34–87) | 68 (45–84) | 0.42 |
| Sex: n (%) |  |  |  |  |
| Male | 132 (72.9%) | 47 (71.2) | 60 (70.6) | 0.93 |
| Female | 49 (27.1%) | 19 (28.8) | 25 (29.4) |  |
| ICIs: n (%) |  |  |  |  |
| Nivolumab | 33 (18.2%) | 8 (12.1) | 21 (24.7) | 0.0089 |
| Pembrolizumab | 128 (70.7%) | 46 (69.7) | 60 (70.6) |  |
| Atezolizumab | 20 (11.1%) | 12 (18.2) | 4 (4.7) |  |
| N of ICIs doses: median(range) | 4 (1–154) | 1.5 (1–2) | 21 (5–154) | <0.0001 |
| RDW: median(range) | 14.6 (12.0–25.1) | 14.8 (12–25.1) | 14.3 (12.1–23.7) | 0.011 |

The p-values were calculated by comparing the short term and long-term groups.

ICIs; Immune Checkpoint Inhibitors, RDW; Red blood cell Distribution Width.

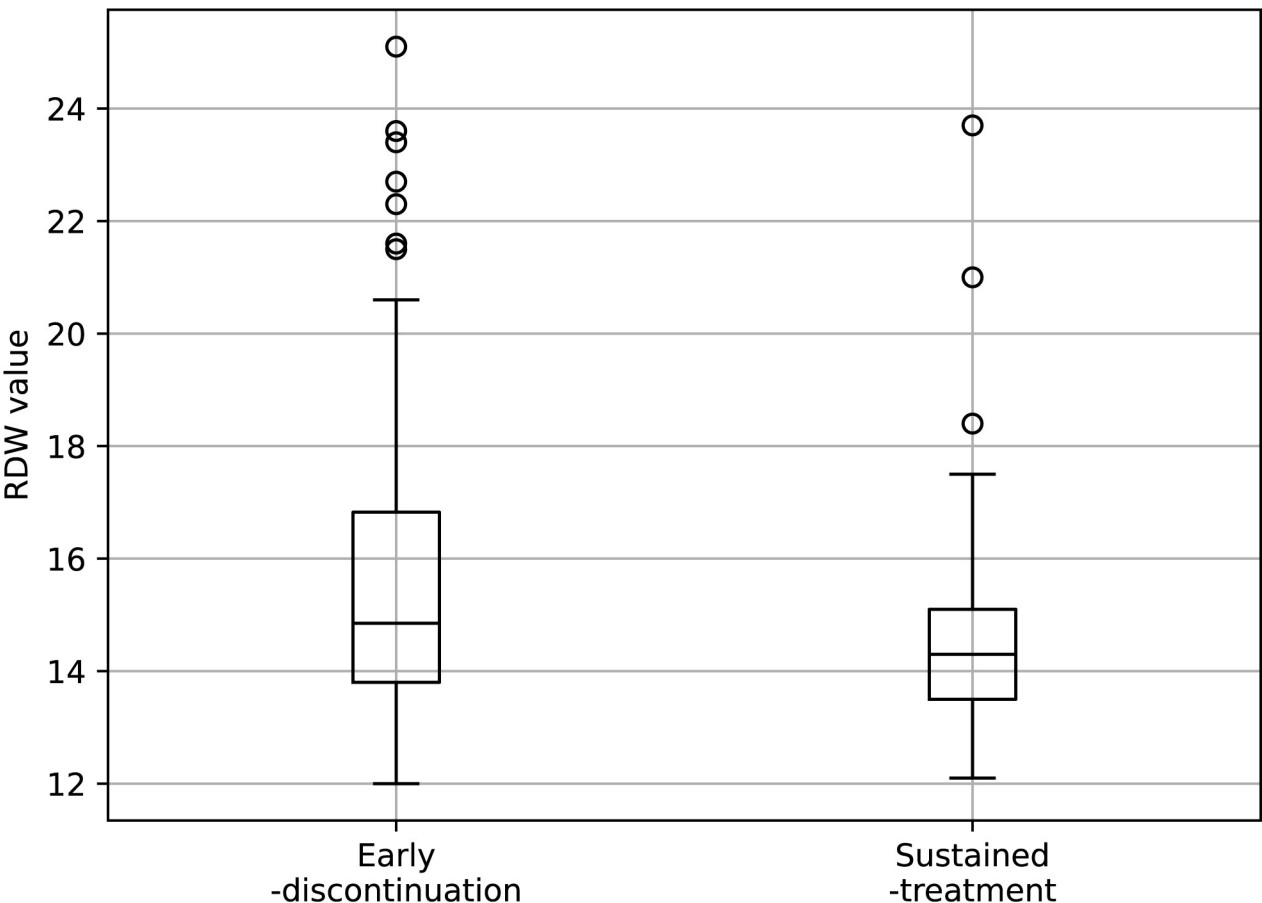

**Fig 5. Box plots of scatter plot of RDW for sustained-treatment and early-discontinuation.** This figure is a boxplot showing the RDW values for the Early-discontinuation and Sustained-treatment groups of ICIs. The RDW values were a median of 14.9 (range 12.0–25.1) for the Early-discontinuation group, and 14.3 (12.1–23.7) for the Sustained-treatment group.

RDW indicates the degree of variation in red blood cell volumes, calculated from a histogram where high RDW signifies greater heterogeneity in cell sizes. Such high RDW arises in conditions like anemia with rapid red blood cell turnover. Elevated RDW has been linked to worse prognosis in cardiovascular diseases including myocardial infarction and heart failure [20–22]. High RDW also associates with stroke severity and outcomes [23]. In this study, higher RDW correlated with fewer ICI administrations, with a regression coefficient of -0.23 (p = 0.0008). This suggests high RDW predicts poorer ICI response. Other studies demonstrate similar connections between high pre-treatment RDW and worse cancer prognosis, including colorectal and metastatic renal cell carcinomas [24,25]. As with these studies, our research suggests that high RDW values are associated with a poor prognosis.

Potential reasons for this relationship include RDW's reflection of inflammation and malnutrition states, which can impair ICI efficacy. The inflammatory cytokine interleukin-6 (IL-6) shows positive correlation with RDW [26,27]. IL-6 may inhibit anti-tumor immunity through effects on CD4+ T cell differentiation and CD8+ T cell cytotoxicity [28–30]. High RDW also associates with poor performance status and malnutrition in lung and esophageal cancers [31,32], both of which negatively impact ICI therapy [33–35].

Validation part 1 verified pembrolizumab administrations as a surrogate for PFS using EMR data. A strong correlation (coefficient 0.80) supported pembrolizumab doses reflecting

PFS, despite discontinuations. Thus, administrations reasonably substituted for PFS. Furthermore, weak negative correlation between RDW and pembrolizumab administrations validated the exploratory results. Validation part 2 extended the RDW-administration association to nivolumab and atezolizumab in NSCLC, using CDW data from two hospitals. Univariate regression demonstrated significant relationship between RDW and administrations (coefficient -0.19, p = 0.0081), confirming pre-treatment RDW predicts decreasing doses. This suggests the RDW-administration link applies beyond pembrolizumab to other ICIs.

The strength of this study is the efficient collection of data from a large number of cases and the use of Python to conduct multiple trials, which facilitated the identification of more reliable predictors. However, there are several limitations. The CDW we used only provided structured data such as birth dates, gender, blood and urine test data, and medication administration dates, so the reasons for medication discontinuation were unclear. Moreover, factors that could significantly influence treatment effectiveness and duration, such as pre-treatment Performance Status (PS) and the presence of adverse events, were unknown and may have affected the results. These limitations could be addressed by external validation data sets or validation with a prospective cohort, which would strengthen the generalizability of the findings.

In conclusion, this study demonstrated that the value of RDW prior to ICI treatment could predict the duration of treatment with ICI as well as PFS. The emerging use of CDW in clinical studies offers a promising opportunity to refine therapeutic strategies, which may contribute to better patient outcomes in the future.

## Supporting information

**S1 Table. Patient characteristics and laboratory data in the exploratory study.** RBC; Red Blood Cell count, Hb; Hemoglobin, Hct; Hematocrit, MCV; Mean Corpuscular Volume, MCH; Mean Corpuscular Hemoglobin, MCHC; Mean Corpuscular Hemoglobin Concentration, RDW; Red Cell Distribution Width, WBC; White Blood Cell count, Neu%; Neutrophil percentage, Neu_seg; Segmented Neutrophils, Neu_stab; Stab Neutrophils, Neu_count; Neutrophil count, Eo%; Eosinophil percentage, Eo_count; Eosinophil count, Baso%; Basophil percentage, Baso_count; Basophil count, Mono%; Monocyte percentage, Mono_count; Monocyte count, Lym%; Lymphocyte percentage, Lym_count; Lymphocyte count, PLT; Platelet count, MPV; Mean Platelet Volume, T-Pro; Total Protein, Alb; Albumin, AST; Aspartate Aminotransferase, ALT; Alanine Aminotransferase, ALP; Alkaline Phosphatase, G-GTP; Gamma-Glutamyl Transferase, T-Bil; Total Bilirubin, CK; Creatine Kinase, LDH; Lactate Dehydrogenase, BUN; Blood Urea Nitrogen, Cre; Creatinine, eGFR; Estimated Glomerular Filtration Rate, Na; Sodium, K; Potassium, Cl; Chloride, Ca; Calcium, CRP; C-Reactive Protein.
(XLSX)

**S2 Table. Patient characteristics and treatment outcomes in the validation part 1.** The parameters are as follows: Sex (M; Male, F; Female), Number of Pembrolizumab doses (N of Pembrolizumab doses), Progression-Free Survival in days (PFS (days)), Age, Pathology (type of cancer, in this case, Adenocarcinoma), and Red Cell Distribution Width (RDW).
(XLSX)

**S3 Table. Patient characteristics and treatment outcomes in the validation part 2.** This table presents the demographic and clinical characteristics of patients included in the validation part 2 of the study on immune checkpoint inhibitors (ICIs). The parameters are as follows: ICIs (type of immune checkpoint inhibitor), Sex (Male, Female), Age, RDW (Red Cell Distribution Width), Target (classification into Early-discontinuation group), and Number of

ICIs doses (N of ICIs doses).
(XLSX)

## Author Contributions

**Conceptualization:** Nobuaki Kobayashi.

**Data curation:** Hiromi Matsumoto, Taichi Fukushima, Nobuaki Kobayashi, Yuuki Higashino, Momo Hirata, Kohei Somekawa.

**Formal analysis:** Hiromi Matsumoto, Nobuaki Kobayashi.

**Funding acquisition:** Nobuaki Kobayashi.

**Investigation:** Hiromi Matsumoto, Taichi Fukushima, Nobuaki Kobayashi, Suguru Muraoka, Yukiko Ohtsu, Ayami Kaneko, Ryo Nagasawa, Sousuke Kubo, Katsushi Tanaka, Kota Murohashi.

**Methodology:** Nobuaki Kobayashi.

**Project administration:** Nobuaki Kobayashi.

**Supervision:** Nobuaki Kobayashi, Hiroaki Fujii, Keisuke Watanabe, Nobuyuki Horita, Yu Hara, Takeshi Kaneko.

**Visualization:** Hiromi Matsumoto.

**Writing – original draft:** Hiromi Matsumoto, Nobuaki Kobayashi.

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
