## [Decision Letter · Decision Letter 0]

9 Jun 2024

PONE-D-24-05510High Red Blood Cell Distribution Width Attenuates the Effectiveness of Immune Checkpoint Inhibitor Therapy: An Exploratory Study Using a Clinical Data WarehousePLOS ONE

Dear Dr. Kobayashi,

Thank you for submitting this manuscript to PLOS ONE. I apologize that its review took some time. Two referees have now commented on the manuscript. They have asked for some clarifications and provided some suggestions. I hope you and other co-authors will be able to address all of them with minor revisions of the manuscript or through rebuttals in a response-to-review document. 

We look forward to receiving your revised manuscript.

Kind regards,

Santosh K. Patnaik, MD, PhD

Academic Editor

PLOS ONE

Journal Requirements:

2. In the online submission form, you indicated that "Data cannot be shared publicly due to ethical considerations. The data underlying the results presented in this study are available from the Yokohama University Ethics Committee. Interested researchers who meet the criteria for access to confidential data can request the data by contacting the corresponding author."

Reviewers' comments:

Reviewer's Responses to Questions

**Comments to the Author**

1. Is the manuscript technically sound, and do the data support the conclusions?

Reviewer #1: Yes

Reviewer #2: Yes

2. Has the statistical analysis been performed appropriately and rigorously? 

Reviewer #1: Yes

Reviewer #2: Yes

3. Have the authors made all data underlying the findings in their manuscript fully available?

Reviewer #1: Yes

Reviewer #2: Yes

4. Is the manuscript presented in an intelligible fashion and written in standard English?

Reviewer #1: Yes

Reviewer #2: Yes

5. Review Comments to the Author

Reviewer #1: I find the manuscript to be very interesting. The search for prognostic biomarkers related to ICI treatment outcomes is now a highly debated topic, especially with the aim of identifying those patients who may not benefit. In this perspective, the work appears to be well-structured, certainly deserving of publication. I found the statistical analyses performed to be comprehensive. They indicate that RDW could play a prognostic role in predicting shorter ICI therapy duration and PFS.

I would like to make some notes: some are of a purely grammatical nature, and others are more significant regarding the statistical analyses.

- Line 4, Short title: I think it is better to use effectiveness than efficacy.

- Line 31-33, Abstract, Background: It should sound better as "This study utilizes a Clinical Data Warehouse (CDW) to explore the prognostic significance of novel blood-based factors, such as the neutrophil-to-lymphocyte ratio and red cell distribution width (RDW), to enhance the prediction of ICI therapy benefit."

- Line 36, Methods: It misses the extended form of "NSCLC" and "EMR".

- Abstract, Methods: You should mention ROC curve analysis.

- Abstract, Results: You should mention the total number of patients or the number of patients in each cohort.

- Line 51, Abstract, Conclusions: It is better to use "suggests" than "affirms"

- Line 52, Abstract, Conclusions: It is better to use "initiation" than "commencement"

- Line 61, Introduction: Remove "EGFR".

- Line 63, Introduction: ".....prognostically significant." Here, it misses the citation.

- Line 69, Introduction: Remove "rate".

- Line 76, Introduction: You should substitute "predicitve" with "prognostic". These terms have different meanings.

- Line 78, Introduction: You can cite "Maffezzoli M, Santoni M, Mazzaschi G, et al. External validation of a red cell-based blood prognostic score in patients with metastatic renal cell carcinoma treated with first-line immunotherapy combinations. Clin Exp Metastasis. 2024;41(2):117-129. doi:10.1007/s10585-024-10266-6."

- Line 92, Introduction: it is better to use effectiveness than efficacy.

- Line 95, Introduction: You should substitute "predicitve" with "prognostic".

- Line 139, Methods: Introduce "NSCLC" as abbreviation (then use it in the other parts of the manuscript).

- Line 150, Statistical Analysis: Better to use "fewer times" than "less"

- Statistical Analysis: the statistical plan of the study is solid and appropriate. The dual validation approach, logistic regression, ROC curve and correlation analyses, as well as the use of CDW, enhance the robustness and generalizability of the findings. However, there are several areas where the methodology could be strengthened:

1) I think that incorporating time-to-event analysis (Cox proportional hazards models) could provide additional insights about the biomarkers and their association with PFS.

2) I think that exploring additional novel biomarkers or combining multiple markers into a composite score might improve predictive accuracy.

- Table 1: Please, clarify whether the age values are reported as median or mean, and explain ranges in brackets.

- Line 181, Exploration Part: Please, specify that there's no differences in the DISTRIBUTION of age, gender, etc. between the two groups.

- Line 195: blood urea nitrogen (BUN).

- Line 196: Chlorine (Cl)

- Line 203: the reasons for selecting specific variables for the multivariate model should be clearly justified.

- Line 206: The multivariate logistic regression includes multiple predictors, but only RDW remained significant. It might be beneficial to explore interactions between variables and other potential confounders.

Line 208: Please, justify the selection of the RDW threshold (15.5) with more detail (how this cutoff was chosen and its clinical relevance).

- Line 246: You should substitute "predicitve" with "prognostic".

- Validation part 2: It might be beneficial to explore interactions between variables and other potential confounders.

- Line 301: use "NSCLC".

- Line 318: You can cite "Maffezzoli M, Santoni M, Mazzaschi G, et al. External validation of a red cell-based blood prognostic score in patients with metastatic renal cell carcinoma treated with first-line immunotherapy combinations. Clin Exp Metastasis. 2024;41(2):117-129. doi:10.1007/s10585-024-10266-6."

- Discussion: External validation using data from different institutions or a prospective cohort would strengthen the findings' generalizability.

Reviewer #2: The manuscript by Kobayashi et al. is an interesting study that leverages an often-overlooked source of data collection: the Clinical Data Warehouse (CDW). This approach is commendable as it opens new avenues for discovering prognostic biomarkers in cancer treatment, an area where current markers exhibit limited predictive accuracy. The strength of the study lies in the rigorous validation process utilizing two different cohorts to ensure the reliability of the exploratory analysis. By confirming findings in both the CDW and EMR cohorts, the study supports the identification of elevated red cell distribution width (RDW) as a potential for predicting the efficacy of immune checkpoint inhibitor (ICI) therapy and paves the way for future studies on the topic. I recommend this article for publication with a few modifications, as outlined in the following suggestions.

• Abstract

o Line 36: Expand EMR when using for the first time.

o Line 36: Expand NSCLC when using for the first time.

o Clarify the methodology in the abstract: Were there two cohorts? Was the study only done on NSCLC patients? Mention the number of patients in each cohort (n=__). Line 41: "... utilized CDW for discovery and EMR/CDW for validating prognostic biomarkers for ICI treatment using the number of doses of ICI as a proxy." Or restructure the abstract to make it clear that the question is about finding biomarkers to predict ICI response, with an intermediate step being finding a PFS proxy.

• Introduction

o Line 63: Provide a citation for immune-related adverse events. Explain the tail effect and early non-response.

o Line 71: Clarify "various measurement methods"—are they non-standardized? Do they give different results?

• Methodology

o State why the authors selected only lung cancer patients for the validation cohorts.

o Explain how the authors selected the number of doses for early-discontinuation and sustained-treatment.

o Mention the statistical test used for group comparisons in the text as well as in the table captions.

• Results

o Mention the criteria used to select variables for the multivariable regression (Line 203).

o State if RDW was directly compared with PFS, not just the number of pembrolizumab doses, in the first validation cohort. Include additional data if there is an association, as it would strengthen the notion that the number of doses is a good surrogate for PFS.

o Compare patient characteristics of the exploratory cohort and the validation cohorts and identify any possible group differences. Add this in the supplementary materials.

o Table 6: Perform a subgroup comparison for nivolumab, pembrolizumab, and atezolizumab (n%) between the early-discontinuation group and the sustained-treatment group instead of using an overall p-value.

• Discussion

o It is interesting and logical that the number of pembrolizumab doses correlates with progression-free survival. Has any other study reported using the number of doses as a PFS proxy?

o Line 329: Clarify that there is a weak negative correlation between RDW and actual PFS. In the results, it is stated as between RDW and the number of pembrolizumab doses.

• The authors should revise the language to improve readability. Here are a couple of examples:

o Line 37: Use "between" instead of "from."

o Use active voice wherever possible. Split sentences into two instead of using long sentences (Line 60-63).

o Line 300: Remove "the three ICIs."

6. PLOS authors have the option to publish the peer review history of their article (what does this mean?). If published, this will include your full peer review and any attached files.

Reviewer #1: No

Reviewer #2: No

---

## [Author Response · Author response to Decision Letter 0]

11 Jul 2024

Response to Reviewers

Manuscript ID: PONE-D-24-05510 

Title: High Red Blood Cell Distribution Width Attenuates the Effectiveness of Immune Checkpoint Inhibitor Therapy: An Exploratory Study Using a Clinical Data Warehouse

Dear Dr. Patnaik,

We sincerely appreciate the time and effort that you and the reviewers have invested in evaluating our manuscript. We have carefully considered all comments and suggestions and have made the necessary revisions to address each point. Our detailed responses to the reviewers' comments are provided below. We believe these revisions have significantly improved the quality and clarity of our manuscript.

Reviewer #1 Comments:

 Short title: 

 Comment: I think it is better to use "effectiveness" than "efficacy".

 Response: The short title has been revised to "High Red Blood Cell Distribution Width Attenuates the Effectiveness of Immune Checkpoint Inhibitor Therapy".

 Comment: It should sound better as "This study utilizes a Clinical Data Warehouse (CDW) to explore the prognostic significance of novel blood-based factors such as the neutrophil-to-lymphocyte ratio and red cell distribution width (RDW) to enhance the prediction of ICI therapy benefit."

 Response: The background section of the abstract has been revised accordingly.

 Comment: It misses the extended form of "NSCLC" and "EMR".

 Response: The full forms "Non-Small Cell Lung Cancer (NSCLC)" and "Electronic Medical Record (EMR)" have been included at their first mention.

Abstract, Methods:

 Comment: You should mention ROC curve analysis.

 Response: ROC curve analysis has been mentioned in the abstract methods section. The sentence has been revised to "Logistic regression identified factors predicting ≤2 or ≥5 pembrolizumab doses as proxies for progression-free survival (PFS), and Receiver Operating Characteristic analysis was used to examine their predictive ability."

Abstract, Results:

 Comment: You should mention the total number of patients or the number of patients in each cohort.

 Response: The total number of patients and the number of patients in each cohort have been added to the abstract results section. The following sentence has been added; “A total of 609 cases (428 in the exploratory cohort and 181 in the validation cohort) from CDW and 44 cases from EMR were selected for study.”

 Comments:

 Line 51: It is better to use "suggests" than "affirms".

 Line 52: It is better to use "initiation" than "commencement".

 Response: The conclusions in the abstract have been revised to use "suggests" and "initiation".

Introduction:

 Comments:

 Line 61: Remove "EGFR".

 Response: We have removed "EGFR" from line 61 in the Introduction.

 Comments: 

 Line 63: ".....prognostically significant." Here it misses the citation.

 Response: We have added a citation for "prognostically significant" on line 63 in the Introduction.

 Comments: 

 Line 69: Remove "rate".

 Response: We have removed it.

 Comments: 

 Line 76: Substitute "predictive" with "prognostic".

 Response: We have substituted "predictive" with "prognostic" on line 95 in the Introduction.

 Comments: 

 Line 78: Cite "Maffezzoli M, et al. External validation of a red cell-based blood prognostic score in patients with metastatic renal cell carcinoma treated with first-line immunotherapy combinations. Clin Exp Metastasis. 2024;41(2):117-129."

 Response: Thank you for your suggestion. We have cited this ariticle.

 Comments: 

 Line 92: Use "effectiveness" than "efficacy".

 Line 95: Substitute "predictive" with "prognostic".

 Response: These revisions have been made in the introduction.

Methods:

 Comment: Line 139: Introduce "NSCLC" as abbreviation.

 Response: "NSCLC" has been introduced as an abbreviation.

 Comments:

 Line 150: Better to use "fewer times" than "less".

 Response: We have changed "less" to "fewer times" on line 150 in the Statistical Analysis section.

 Comments: 

 Statistical Analysis: the statistical plan of the study is solid and appropriate. The dual validation approach, logistic regression, ROC curve and correlation analyses, as well as the use of CDW, enhance the robustness and generalizability of the findings. However, there are several areas where the methodology could be strengthened:

1) I think that incorporating time-to-event analysis (Cox proportional hazards models) could provide additional insights about the biomarkers and their association with PFS.

 Line 150: Incorporating time-to-event analysis (Cox proportional hazards models) could provide additional insights.

 Response: We appreciate the positive feedback regarding our statistical plan and the robustness of our methodology. Regarding your suggestion to incorporate time-to-event analysis using Cox proportional hazards models, we acknowledge the value of this approach. However, as described in the limitations section of our manuscript, the Clinical Data Warehouse (CDW) utilized in our study does not provide information on whether an event (progressive disease, PD, in this case) occurred or not. Consequently, conducting a Cox regression analysis was not feasible. We have now revised the relevant section of our manuscript to clarify this point further:

 "Logistic regression identified factors predicting ≤2 or ≥5 pembrolizumab doses as proxies for progression-free survival (PFS), and Receiver Operating Characteristic analysis was used to examine their predictive ability."

2) I think that exploring additional novel biomarkers or combining multiple markers into a composite score might improve predictive accuracy.

 Response: In this study, we examined all possible candidate biomarkers available from the Data Warehouse (DWH). However, RDW was the only factor that remained independent and significant, even in multivariate analysis. This finding underscores the robustness of RDW as a prognostic biomarker within the constraints of the data available to us.

Comments:

 Table 1: Please, clarify whether the age values are reported as median or mean, and explain ranges in brackets.

 Response: We have clarified in Table 1 that age values are reported as median and explained the ranges in brackets.

Comments:

 Line 181, Exploration Part: Please, specify that there's no differences in the DISTRIBUTION of age, gender, etc. between the two groups.

 Response: We have revised the manuscript to specify that there were no significant differences in the distribution of age, gender, or the department in which the ICI was administered between the two groups. The added sentence is: "There were no significant differences in age, gender, or the department in which the ICI was administered in the two groups."

Comments:

 Line 195: blood urea nitrogen (BUN).

 Line 196: Chlorine (Cl)

 Response: We have added the full forms for BUN and Cl on lines 195 and 196, respectively.

Comments:

 Line 203: The reasons for selecting specific variables for the multivariate model should be clearly justified.

 Response: We have revised the manuscript to clarify the selection process for the variables in the multivariate model. The added sentence is: "Of these 15 items, 5,005 models were created with 9 items as explanatory variables, and the explanatory variables of the model with the best classification performance were selected." in line 235 of revised manuscript. 

 Comments:

 Line 206: The multivariate logistic regression includes multiple predictors, but only RDW remained significant. It might be beneficial to explore interactions between variables and other potential confounders.

 Response: Among the variables used in the multivariate analysis in this study, Hb and Hct and WBC and neutrophil count have strong correlations with each other and may be multicollinear. However, there is no item with a strong correlation for RDW, and since the subsequent validation also found an association with prognosis, we do not believe that multicollinearity has a significant impact on the present results.

Comments:

 Line 208: Justify the selection of the RDW threshold (15.5) with more detail.

 Response: We have revised the manuscript to provide a detailed justification for the selection of the RDW threshold. The RDW threshold value of 15.5 was obtained using the Youden Index, which is described in the Methods section. This method was used to determine the optimal cutoff point for distinguishing between early discontinuation and sustained treatment groups based on RDW values.

Revised Sentence in Manuscript:

"For this model, the only variable that was significant was RDW (p=0.0008). In the ROC analysis using sustained-treatment and early-discontinuation with RDW, the Area Under the Curve was 0.60, and at an RDW value of 15.5, the sensitivity for detecting the early-discontinuation administration group was 0.41 and the specificity was 0.79 (Fig 2). This value was obtained using the Youden Index (described in Methods)."

Comment: 

 Validation part 2: It might be beneficial to explore interactions between variables and other potential confounders.

 Response: Since this validation part only examined the validity of the factors identified in the exploratory part as prognostic predictors of RDW, other factors were not examined.

Comment: 

 Line 301: Use "NSCLC".

 Response: "NSCLC" has been used accordingly.

Comment:

 Line 318: Cite "Maffezzoli M, et al. External validation of a red cell-based blood prognostic score in patients with metastatic renal cell carcinoma treated with first-line immunotherapy combinations. Clin Exp Metastasis. 2024;41(2):117-129."

 Response: Thank you for your suggestion. This citation has been added.

Discussion:

 Comment: External validation using data from different institutions or a prospective cohort would strengthen the findings' generalizability.

 Response: We acknowledge the importance of external validation for strengthening the generalizability of our findings. In the revised manuscript, we have addressed this point as follows: "These limitations could be addressed by external validation data sets or validation with a prospective cohort, which would strengthen the generalizability of the findings."

Reviewer #2 Comments:

 Comments:

• Abstract

 Line 36: Expand EMR when using for the first time.

 Response: The term "Electronic Medical Record (EMR)" has been expanded at its first occurrence in the manuscript.

 Comments:

 Line 36: Expand NSCLC when using for the first time.

 Response: The term "Non-Small Cell Lung Cancer (NSCLC)" has been expanded at its first occurrence in the manuscript.

 Comments:

 Clarify the methodology in the abstract: Were there two cohorts? Was the study only done on NSCLC patients? Mention the number of patients in each cohort (n=__). Line 41: "... utilized CDW for discovery and EMR/CDW for validating prognostic biomarkers for ICI treatment using the number of doses of ICI as a proxy." Or restructure the abstract to make it clear that the question is about finding biomarkers to predict ICI response, with an intermediate step being finding a PFS proxy.

 Response 1: The methodology in the abstract has been clarified to specify the cohorts and the patient numbers. The revised abstract reads:

"This retrospective study utilized a CDW to explore factors associated with pembrolizumab treatment duration validated in non-small cell lung cancer (NSCLC) patient cohorts from electronic medical records (EMR) and CDW. The CDW contained anonymized data on demographics, diagnoses, medications, and tests for cancer patients treated with ICIs between 2017-2022. Logistic regression identified factors predicting ≤2 or ≥5 pembrolizumab doses as proxies for progression-free survival (PFS), and Receiver Operating Characteristic analysis was used to examine their predictive ability. These factors were validated by correlating doses with PFS in the EMR cohort and re-testing their significance in the CDW cohort with other ICIs. This dual approach utilized the CDW for discovery and EMR/CDW cohorts for validating prognostic biomarkers before ICI treatment."

 Response 2: The number of patients in each cohort has been mentioned: "A total of 609 cases (428 in the exploratory cohort and 181 in the validation cohort) from CDW and 44 cases from EMR were selected for study."

 Comments:

• Introduction

 Line 63: Provide a citation for immune-related adverse events. Explain the tail effect and early non-response.

 Response: We have revised the manuscript to include a citation for immune-related adverse events and to explain the tail effect and early non-response. The revised sentence reads: "Compared to traditional cytotoxic chemotherapeutics and molecular targeted therapies like tyrosine kinase inhibitors, ICIs have distinctive features, including long-lasting anti-tumor effects (a long tail effect), cancer progression early after initiation of treatment (early non-response)[4], and the occurrence of adverse events similar to autoimmune diseases mediated by the immune system[5], some of which can be severe and prognostically significant."

 Comments:

 Line 71: Clarify "various measurement methods".

 Response: We appreciate the reviewer's comment on clarifying the "various measurement methods." We have revised this section to more accurately reflect our intended meaning. In the revised manuscript:

"Biomarkers such as tumor PD-L1 expression rate, microsatellite instability, and Tumor Mutation Burden are currently used to predict the effects of ICIs. However, these biomarkers have issues like insufficient predictive accuracy, variability in assessment methods (including different antibodies used for PD-L1 evaluation and potential inter-observer variability), and heterogeneity within the same tumor[7, 8]. Therefore, there is a demand for the development of biomarkers that can more accurately predict the therapeutic effects of ICIs."

 Methodology:

 Comments:

 State why the authors selected only lung cancer patients for the validation cohorts.

 Response: In the exploratory part of the study, we analyzed all patients who received ICI regardless of the type of cancer in order to ensure the number of cases. However, the prognosis of each cancer type is different, and it is desirable to analyze each type of cancer separately. Therefore, in the exploratory part, potential prognostic factors were identified, and in the validation part, whether the factors obtained in the exploratory part had the same predictive ability for a single cancer type was verified.

 Comments:

 Explain how the authors selected the number of doses for early-discontinuation and sustained-treatment.

 Response: We have revised the manuscript to clarify the selection of the number of doses for early-discontinuation and sustained-treatment. The early-discontinuation group was defined as receiving two or fewer doses, and the sustained-treatment group was defined as receiving five or more doses. This threshold was selected by reference to the overall distribution and to ensure a sufficient number of cases for statistical analysis. Cases where the period between the first dose and the last day of CDW storage was less than 12 weeks were excluded as it was impossible to determine whether they belonged to the sustained-treatment group or not. This is detailed in the revised manuscript.

Revised Section in Manuscript:

"Early-discontinuation group was defined as two or fewer doses and sustained-treatment was defined as five or more doses. The analysis included 320 subjects. Cases in which the period between the first dose and the last day of CDW storage was less than 12 weeks were excluded because it was impossible to determine whether they belonged to the sustained-treatment group or not. There were 99 cases in the early-discontinuation group and 221 cases in the sustained-treatment group."

 Comments:

 Mention the statistical test used for group comparisons in the text as well as in the table captions.

 Response: We have revised the manuscript to mention the statistical tests used for group comparisons in both the text and the table captions. For example, in Table 1, we stated that the p-values were calculated by compa

---

## [Editor Report · Decision Letter 1]

16 Jul 2024

High Red Blood Cell Distribution Width Attenuates the Effectiveness of Immune Checkpoint Inhibitor Therapy: An Exploratory Study Using a Clinical Data Warehouse

PONE-D-24-05510R1

Dear Dr. Kobayashi,

We received the revised version of this manuscript. I have examined it and the accompanying response to review letter. The revision appropriately addresses all concerns that were raised by the reviewers.

I am therefore pleased to inform you that your manuscript has been judged scientifically suitable for publication and will be formally accepted for publication once it meets all outstanding technical requirements.

The journal does not perform copy editing, so the publication will be entirely based on the revised manuscript. I recommend that you and your colleagues take a final look to correct any language-related or other error that may have gone unnoticed.

Kind regards,

Santosh K. Patnaik, MD, PhD

Academic Editor

PLOS ONE

---

## [Editor Report · Acceptance letter]

24 Jul 2024

PONE-D-24-05510R1 

PLOS ONE

Dear Dr. Kobayashi, 

I'm pleased to inform you that your manuscript has been deemed suitable for publication in PLOS ONE. Congratulations! Your manuscript is now being handed over to our production team.

Kind regards, 

on behalf of

Dr. Santosh K. Patnaik 

Academic Editor

PLOS ONE